# Multimodal Unsupervised Speech Translation for Recognizing and Evaluating Second Language Speech

**Yun Kyung Lee** * and **Jeon Gue Park**

Artificial Intelligence Research Laboratory, Electronics and Telecommunications Research Institute (ETRI), Daejeon 34129, Korea; jgp@etri.re.kr
* Correspondence: yunklee@etri.re.kr

**Abstract:** This paper addresses an automatic proficiency evaluation and speech recognition for second language (L2) speech. The proposed method recognizes the speech uttered by the L2 speaker, measures a variety of fluency scores, and evaluates the proficiency of the speaker's spoken English. Stress and rhythm scores are one of the important factors used to evaluate fluency in spoken English and are computed by comparing the stress patterns and the rhythm distributions to those of native speakers. In order to compute the stress and rhythm scores even when the phonemic sequence of the L2 speaker's English sentence is different from the native speaker's one, we align the phonemic sequences based on a dynamic time-warping approach. We also improve the performance of the speech recognition system for non-native speakers and compute fluency features more accurately by augmenting the non-native training dataset and training an acoustic model with the augmented dataset. In this work, we augment the non-native speech by converting some speech signal characteristics (style) while preserving its linguistic information. The proposed variational autoencoder (VAE)-based speech conversion network trains the conversion model by decomposing the spectral features of the speech into a speaker-invariant content factor and a speaker-specific style factor to estimate diverse and robust speech styles. Experimental results show that the proposed method effectively measures the fluency scores and generates diverse output signals. Also, in the proficiency evaluation and speech recognition tests, the proposed method improves the proficiency score performance and speech recognition accuracy for all proficiency areas compared to a method employing conventional acoustic models.

**Keywords:** fluency evaluation; speech recognition; data augmentation; variational autoencoder; speech conversion

## 1. Introduction

As the demand for untact technology in various fields increases and machine learning technologies advance, the need for computer-assisted second language (L2) learning contents has increased [1–4]. The widely used method for learning a second language is to practice listening, repeating, and speaking language. A GenieTutor, one of the second language (English at present) systems, plays the role of a language tutor by asking questions to the learners, recognizing their speech, which is answered in second language, checking grammatical errors, evaluating the learners' spoken English proficiency, and providing feedbacks to help L2 learners practice their English proficiency. The system comprises several topics, and the learners can select a topic to have communication with the system based on the role-play scenarios. After the learner finishes the speaking of each sentence, the system measures various fluency factors such as pronunciation score, word score, grammar error, stress pattern, and intonation curve, and provides feedback to learners about them compared with the fluency factors of the native speakers [5–7].

The stress and rhythm scores are one of the important factors for fluency evaluation in English speaking, and they are computed by comparing the stress pattern and the rhythm

distribution of the L2 speaker with those of native speakers. However, in some cases, the phonemic sequences of speeches uttered by the L2 speaker and the native speaker are recognized differently according to the pronunciation of the learner. Learners may mispronounce or pronounce different words from the referred one. In such cases, the stress and rhythm scores cannot be computed using the previous pattern comparison methods [8–14].

In order to solve this problem, we proposed a dynamic time-warping (DTW)-based stress and rhythm scores measurement method. We aligned the learner's phonemic sequence with the native speaker's phonemic sequence by using the DTW approach, and then computed the stress patterns and rhythm distributions from the aligned phonemic sequences [15–19]. By using the aligned phonemic sequences, we detected the learner's pronunciation error phonemes and computed an error-tagged stress pattern and scores which are deducted by the presence of error phonemes if there is an erroneous phoneme. We computed two stress scores: a word stress score and a sentence stress score. The word stress score is measured by comparing the stress patterns of the content words, and the sentence stress score was computed for the entire sentence. The rhythm scores are measured by computing the mean and standard deviation of the time distances between stressed phonemes. Two stress scores and rhythm scores are used to evaluate the English-speaking proficiency of the learner with other fluency features.

The proposed method uses an automatic speech recognition (ASR) system to recognize the speech uttered by the L2 learner and perform a forced-alignment to obtain the time-aligned phonemic sequences. Deep learning has been applied successfully to ASR systems by relying on hierarchical representations that are commonly learned with a large amount of training data. However, non-native speakers' speech significantly degrades the performance of the ASR due to the pronunciation variability in non-native speech, and it is difficult to collect enough non-native data to train. For better performance of the ASR for non-native speakers, we augment the non-native training speech dataset by using a variational autoencoder-based speech conversion model and train an acoustic model (AM) with the augmented training dataset. Data augmentation has been proposed as a method to generate additional training data, increase the quantity of training data, and reduce overfitting for ASR systems [20–25].

The speech conversion (SC) technique is to convert the speech signal from a source domain to that of a target domain, while preserving its linguistic content information. Variational autoencoder (VAE) is a widely used method for speech modeling and conversion. In the VAE framework, the spectral features of the speech are encoded to a speaker-independent latent variable space. After sampling from the latent variable space, sampled features are decoded back to the speaker-dependent spectral features. A conditional VAE, one of the VAE-based speech conversion methods, employs speaker identity information to feed the decoder with the sampled features. With this conditional decoder, the VAE framework can reconstruct or convert input speech by choosing speaker identity information [26–33]. Recently, generative adversarial networks (GANs)-based SC and some frameworks that jointly train a VAE and GAN were proposed [34–44]. However, most conversion frameworks usually assume and learn a deterministic or unimodal mapping, so their significant limitation is the lack of diversity in the converted outputs.

We build up the proposed conversion model based on the VAE, due to its potential in employing latent space to represent hidden aspects of speech signal. In order to improve speech conversion without conditional information and learn more meaningful speaker characteristic information, we proposed a VAE-based multimodal unsupervised speech conversion method. In the proposed method, we assume that the spectral features of speech are divided into a speaker-invariant content factor (phonetic information in speech) and a speaker-specific style factor [45–47]. We employ a single shared content encoder network and an individual style encoder network for each speaker to train the encoder models robustly. The encoded content factor is fed into a decoder with a target style factor to generate converted spectral features. By sampling different style factors, the proposed

model is able to generate diverse and multimodal outputs. In addition, we train our speech conversion model from nonparallel data because parallel data of the source and target speakers are not available in most practical applications and it is difficult to collect such data. By transferring some speech characteristics and converting the speech, we generate additional training data with nonparallel data and train the AM with the augmented training dataset.

We evaluated the proposed method on the corpus of English read speech for the spoken English proficiency assessment [48]. In our experiments, we evaluated the fluency scoring ability of the proposed method by measuring fluency scores and comparing them with the fluency scores of native speakers, and the results demonstrate that the proposed DTW-based fluency scoring method can compute stress patterns and measure stress and rhythm scores effectively even if there are pronunciation errors in the learner's utterances. The spectral feature-related outputs demonstrate that the proposed conversion model can efficiently generate diverse signals while keeping the linguistic information of the original signal. Proficiency evaluation test and speech recognition results with and without an augmented speech dataset also show that the data augmentation with the proposed speech conversion model contributed to improving speech recognition accuracy and proficiency evaluation performance compared to a method employing conventional AMs.

The remainder of this paper is organized as follows. Section 2 briefly describes the second language learning system used in this work. Section 3 describes a description of the proposed DTW-based fluency scoring and VAE-based nonparallel speech conversion method. In Section 4, experimental results are reported, and finally, we conclude and discuss this paper in Section 5.

## 2. Previous Work

GenieTutor is a computer-assisted second language (English at present) learning system. In order to help learners practice their English proficiency, the system recognizes the learners' spoken English responses for given questions, checks content properness, automatically checks and corrects grammatical errors, evaluates spoken English proficiency, and provides educational feedback to learners. Figure 1 shows the schematic diagram of the system [7].

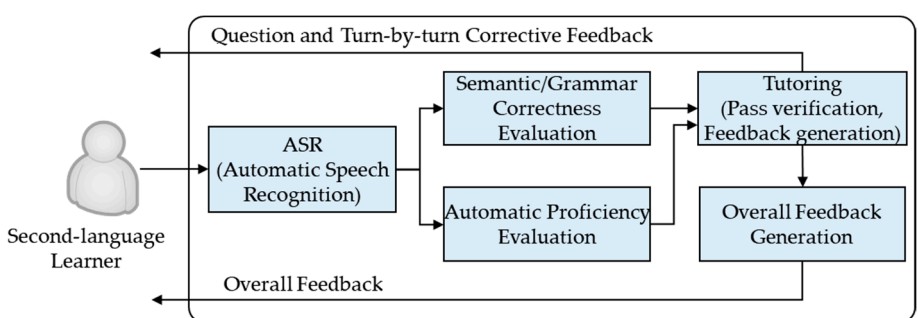

**Figure 1.** Schematic diagram of the GenieTutor system.

The system comprises two learning stages: Think&Talk and Look&Talk. The Think&Talk stage has various subjects, and each subject comprises several fixed role-play dialogues. In this stage, an English learner can select a study topic and a preferred scenario, and then talk with the system based on the selected role-play scenario. After the learner's spoken English response for each given question is completed, the system computes an intonation curve, a sentence stress pattern, and word pronunciation scores. The learner's and a native speaker's intonation curve patterns are plotted as a graph, and the stress patterns of the learner and native speaker are plotted by circles with different sizes below the corresponding word to represent the intensity of each word at a sentence stress level. Once the learner has finished all conversations on the selected subject or all descriptions of

the selected picture, the system semantically and grammatically evaluates the responses, and provides overall feedback.

Figure 2 shows an example of a role-play scenario and educational overall feedback with the system. In the Look&Talk stage, the English learner can select a picture and then describe the selected picture to the system.

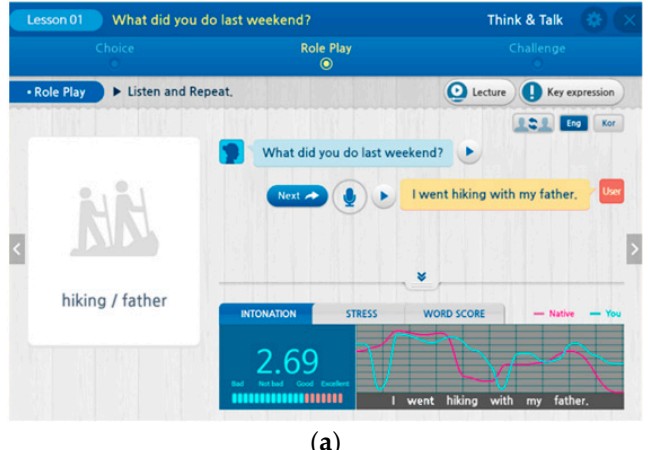
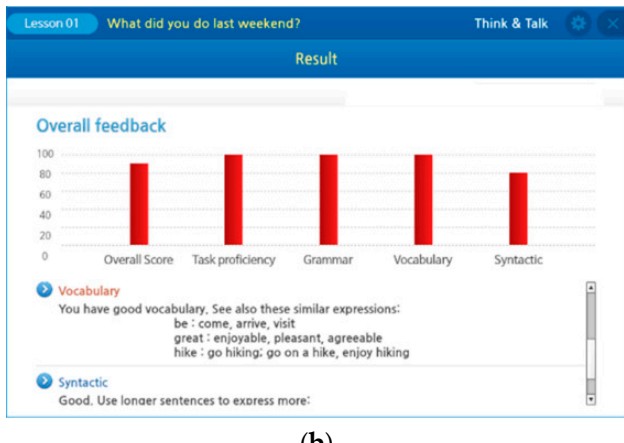

(**a**)  (**b**)

**Figure 2.** Example of a role-play scenario and the overall feedback. (**a**) Example of a role-play dialogue exercise and intonation feedback of the learner with the native speaker. (**b**) Overall feedback of the role-play dialogue exercise.

## 3. Proposed Fluency Scoring and Automatic Proficiency Evaluation Method

Proficiency evaluation with the proposed method consists of fluency features extraction for scoring each proficiency area, proficiency evaluation model training with fluency features, and automatic evaluation of pronunciation proficiency. The proposed method computes various acoustic features, such as speech rate, intonation, and segmental features, from spoken English uttered by non-native speakers according to a rubric designed to evaluate pronunciation proficiency. In order to compute the fluency features, speech signals are recognized using the automatic speech recognition system and time-aligned sequences of words and phonemes are computed using a forced-alignment algorithm. Each time-aligned sequence contains start and end times for each word and phoneme and acoustic scores. Using the time-aligned sequences, the fluency features are extracted in various aspects of each word and sentence. Proficiency evaluation models are trained using the extracted fluency features and scores from human expert raters, and proficiency scores are computed using the fluency features and scoring models. Figure 3 shows a block diagram of the proficiency evaluation model training and evaluating system for automatic proficiency evaluation.

### 3.1. DTW-Based Feature Extraction for Fluency Scoring

Most language learning systems evaluate and score the learners' spoken English compared to the native speaker's one. However, in realistic speaking situations, a learner's English pronunciation often differs from that of native speakers. For example, learners may pronounce given words incorrectly or pronounce different words from the referred one. In such cases, some fluency features, especially stress and rhythm scores, cannot be measured using previous pattern comparison methods. To solve this problem and measure more meaningful scores, the proposed method aligns the phonemic sequence of the sentence uttered by the learner with the native speaker's phonemic sequence through dynamic time-warping (DTW) alignment and computes the stress patterns, stress scores, and rhythm scores from the aligned phonemic sequences. Figure 4 shows a block diagram of the proposed DTW-based stress and rhythm scoring method.

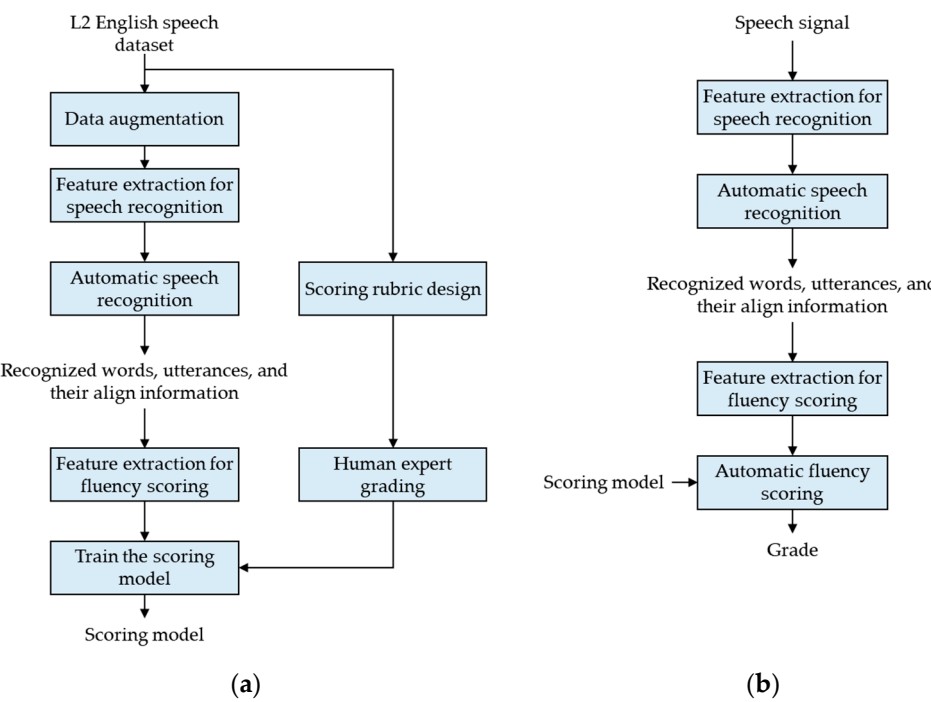

**Figure 3.** Block diagram of the proficiency evaluation model training and evaluating system for automatic proficiency evaluation. (**a**) Flow of the proficiency evaluation model training. (**b**) Flow of the automatic proficiency evaluation.

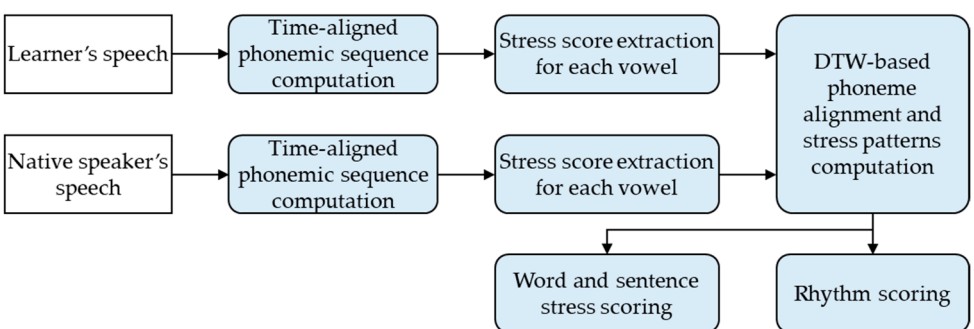

**Figure 4.** Block diagram of the proposed dynamic time-warping (DTW)-based stress and rhythm scoring method.

### 3.1.1. DTW-Based Phoneme Alignment

Dynamic time-warping is a well-known technique for finding an optimal alignment between two time-dependent sequences by comparing them [15]. To compare and align two phonemic sequences uttered by learner and native speaker (reference), we compute a local cost matrix of two phonemic sequences defined by the Euclidean distance. Typically, if a learner and a native speaker are similar to each other, the local cost matrix is small, and otherwise, the local cost matrix is large. The total cost of an alignment path between the learner's and native speaker's phonemic sequences is obtained by summing the local cost measurement values for each pair of elements in two sequences. An optimal alignment path is the alignment path having minimal total cost among all possible alignment paths, and the goal is to find the optimal alignment path and align between two phonemic sequences with the minimal overall cost.

Figure 5 shows an example of DTW-based phonemic sequence alignment and stress patterns' computation results. Error phonemes caused by phoneme mismatch are marked in red, and the stress value of error phonemes was set to 3, which is not a standard stress

value indicated by 0 (no stress), 1 (secondary stress), or 2 (primary stress) in order to compute the deducted stress score according to the presence of the phonemic errors. By tagging error phonemes, the proposed method evaluates the learner's utterance more accurately and helps L2 learners practice their English pronunciation.

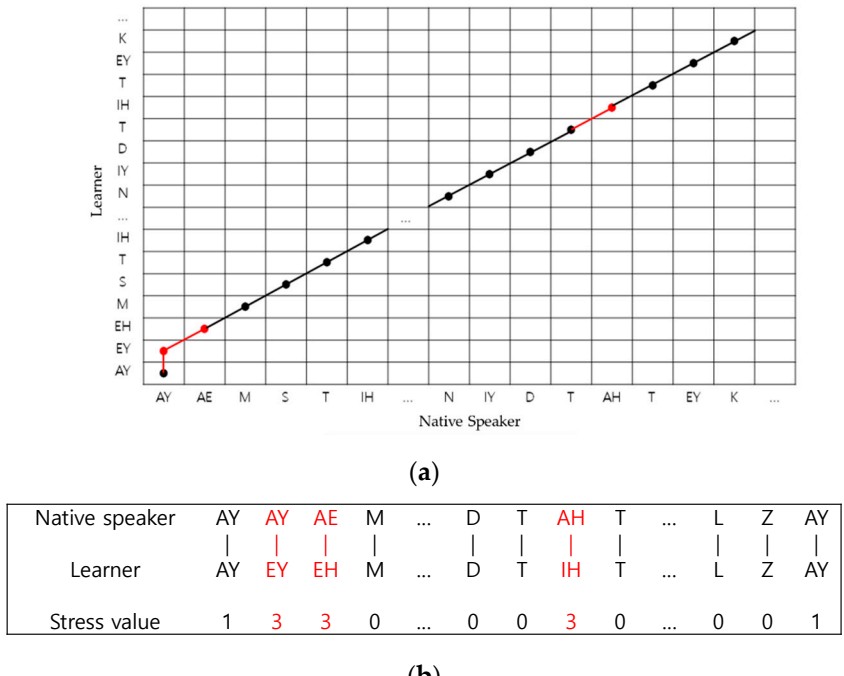

(**a**)

| Native speaker | AY | AY | AE | M | ... | D | T | AH | T | ... | L | Z | AY |
|---|---|---|---|---|---|---|---|---|---|---|---|---|---|
| | \| | \| | \| | \| | | \| | \| | \| | \| | | \| | \| | \| |
| Learner | AY | EY | EH | M | ... | D | T | IH | T | ... | L | Z | AY |
| | | | | | | | | | | | | | |
| Stress value | 1 | 3 | 3 | 0 | ... | 0 | 0 | 3 | 0 | ... | 0 | 0 | 1 |

(**b**)

**Figure 5.** Example of DTW-based phonemic sequence alignment in sentence "I am still very sick. I need to take some pills." (**a**) Aligned alignment path of the phonemic sequences. (**b**) Aligned phonemic sequence and stress values.

### 3.1.2. Stress and Rhythm Scoring

Given the aligned phonemic sequences, the proposed method computes the word stress score, sentence stress score, and rhythm scores. In order to measure the word and sentence stress scores, word stress patterns are computed for each content word in the given sentence, and sentence stress patterns are computed for the entire sentence. Then, the word and sentence stress scores are measured by computing the similarity between the learner's stress patterns and the native speaker's stress patterns.

The rhythm scores are measured by computing the mean and standard deviation of the time intervals between the stressed phonemes. An example of computing the rhythm score in the sentence "I am still very sick. I need to take some pills." is as follows:

- Compute the stress patterns from the aligned phonemic sequences. Table 1 shows an example of the sentence stress pattern. The start times of the stressed phonemes, including the start and end times of the sentence, are highlighted (bold in Table 1) to compute the rhythm features.
- Select the stressed phonemes (highlighted point in Table 1), and compute the mean and standard deviation of the time intervals between them:
  - (1)     Mean time interval: $(0.2 + 0.2 + 0.6 + 0.5 + 1.5)/5 = 0.6$
  - (2)     Standard deviation of the time interval: $(0.16 + 0.16 + 0.0 + 0.01 + 0.81)/5 = 0.23$

**Table 1.** Sentence stress pattern in the sentence "I am still very sick. I need to take some pills."

| Word | Start Time | End Time | Stress Value |
|---|---|---|---|
| I | **0.0** | 0.2 | 1 |
| am | **0.2** | 0.4 | 1 |
| still | **0.4** | 0.8 | 2 |
| very | 0.8 | 1.0 | 0 |
| sick | **1.0** | 1.3 | 1 |
| I | 1.4 | 1.5 | 0 |
| need | **1.5** | 1.8 | 1 |
| to | 1.8 | 2.0 | 0 |
| take | 2.0 | 2.2 | 0 |
| some | 2.2 | 2.5 | 0 |
| pills | 2.5 | **3.0** | 0 |

Table 2 shows an example of the mean values of the mean time interval and the standard deviation of the time interval for each pronunciation proficiency level evaluated by human raters. Proficiency scores 1, 2, 3, 4, and 5 indicate very poor, poor, acceptable, good, and perfect, respectively. As shown in Table 2, the lower the proficiency level, the greater the mean and standard deviation values of the time intervals between the stressed phonemes. Two stress scores and rhythm scores are used for spoken English proficiency evaluation with other features.

**Table 2.** Example of the mean rhythm scores.

| Proficiency Score | Mean Time Interval | Standard Deviation of the Time Interval |
|---|---|---|
| 1 | 1.12 | 0.62 |
| 2 | 0.82 | 0.42 |
| 3 | 0.68 | 0.34 |
| 4 | 058 | 0.32 |
| 5 | 0.56 | 0.31 |

### 3.2. Automatic Proficiency Evaluation with Data Augmentation

The speech recognition system is optimized for non-native speakers as well as natives for educational purposes and smooth interaction. Speech features for computing fluency scores are extracted and decoded into time-aligned sequences by forced-alignment using the non-native acoustic model (AM). In addition, multiple AM scores are used to evaluate proficiency. In order to improve speech recognition accuracy and time-alignment performance, and to compute AM scores more accurately and meaningfully, we augment the training speech dataset and train non-native AM using the augmented training dataset.

In this work, we convert some speech characteristics (style) to generate speech data for augmentation. In the proposed speech conversion model, we assume that each spectral feature of the speech signal is decomposed into a speaker-independent content factor desired to be maintained and each speaker-specific style factor we want to change in latent space. After extracting the content factor from the source speech signal, the proposed conversion model converts the source speech to the desired speech style by extracting the style factor of target speech and recombining it with the extracted content factor. By simply choosing the style factor for this recombination as a source style factor or target style factor, the conversion model can reconstruct or convert speech:

$$\hat{x}_{s \to s} = D\big(E_c(x_s), E_s^s(x_s)\big), \tag{1}$$

$$\hat{x}_{s \to t} = D\big(E_c(x_s), E_s^t(x_t)\big), \tag{2}$$

where $\hat{x}_{s \to s}$ and $\hat{x}_{s \to t}$ are the reconstructed and converted spectra, $x_s$ and $x_t$ are the source and target speech spectra, $D$ is the decoder, and $E_c$, $E_s^s$, and $E_s^t$ denote the content encoder, source style encoder, and target style encoder, respectively. The content encoder network

is shared across both speakers, and the style encoder networks are domain-specific networks for individual speakers. Figure 6 shows a block diagram of the proposed speech conversion method.

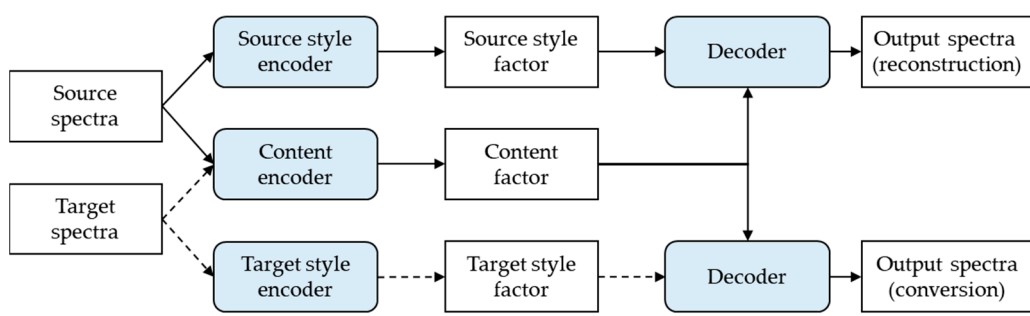

**Figure 6.** Flow of the proposed variational autoencoder (VAE)-based nonparallel speech conversion method. Each of the decoder networks are shared. The arrows indicate flow of the proposed method related to the source spectra or common elements (e.g., source style factor, content factor), and the dashed arrows indicate the flow of the method belonging to the target spectra (e.g., target spectra, target style factor).

As shown in Figure 6, the content encoder network extracts the content factor and is shared across all domains. All convolutional layers of the content encoder were followed by instance normalization (IN) to remove the speech style information and learn domain-independent content information (phoneme in speech). The style encoder network computes the domain-specific style factor for each domain and is composed of multiple separate style encoders (source style encoder and target style encoder in Figure 6) for individual domains. In the style encoders, IN was not used, because it removes the speech style information.

We jointly train the encoders and decoder with multiple losses. To keep encoder and decoder as inverse operations and ensure the proposed system should be able to reconstruct the input spectral features after encoding and decoding, we consider reconstruction loss as follows:

$$L_{recon_s} = \mathrm{E}_{s \sim p(s)}[\|D(E_c(x_s), E_s^s(x_s)) - x_s\|_1]. \tag{3}$$

For the content factor and style factors, we apply a semi-cycle loss in latent variable → speech spectra → latent variable coding direction as the latent space is partially shared. Here, a content reconstruction loss encourages the translated content latent factor to preserve the semantic content information of input spectral features, and a style reconstruction loss encourages style latent factors to extract and change speaker-specific speaking style information. Two semi-cycle losses for source speech are computed as follows:

$$L_{recon_s}^c = \mathrm{E}_{c \sim p(c), s_t \sim q(s_t)}[\|E_c(D(c, s_t)) - c\|], \tag{4}$$

$$L_{recon_s}^s = \mathrm{E}_{c \sim p(c), s_t \sim q(s_t)}[\|E_s^t(D(c, s_t)) - s_t\|], \tag{5}$$

where $c$ denotes the content factor, and $s_t$ and $s_s$ denote the target style factor and source style factor, respectively. The losses for target speech are similarly computed. The full loss of the proposed speech conversion method is the weighted sum of all losses, which is defined as follows:

$$L_{VAE} = \lambda_1(L_{recon_s} + L_{recon_t}) + \lambda_2(L_{recon_s}^c + L_{recon_t}^c) + \lambda_3(L_{recon_s}^s + L_{recon_t}^s), \tag{6}$$

where $\lambda_1$, $\lambda_2$, and $\lambda_3$ control the weights of the components.

## 4. Experimental Results

### 4.1. Dialogue Exercise Result

To validate the effectiveness of the proposed method, we performed computer-assisted fluency scoring experiments with spoken English sentences collected in dialogue scenarios of the GenieTutor system. Figure 7 shows an example of a role-play scenario and fluency scores feedback with the proposed method. Once the learner completes a sentence utterance, the system computes several aspects of pronunciation evaluation and displays them in diagram forms. Learners can check their fluency scores by selecting the sentences they want to check. Learners are provided with overall feedback after finishing all conversations. As shown in Figure 7, the proposed method can efficiently compute the intonation curves and stress patterns of the sentences uttered by the learner even when pronunciation errors occur. In addition, the error words are marked in red, so the learner can see the error parts.

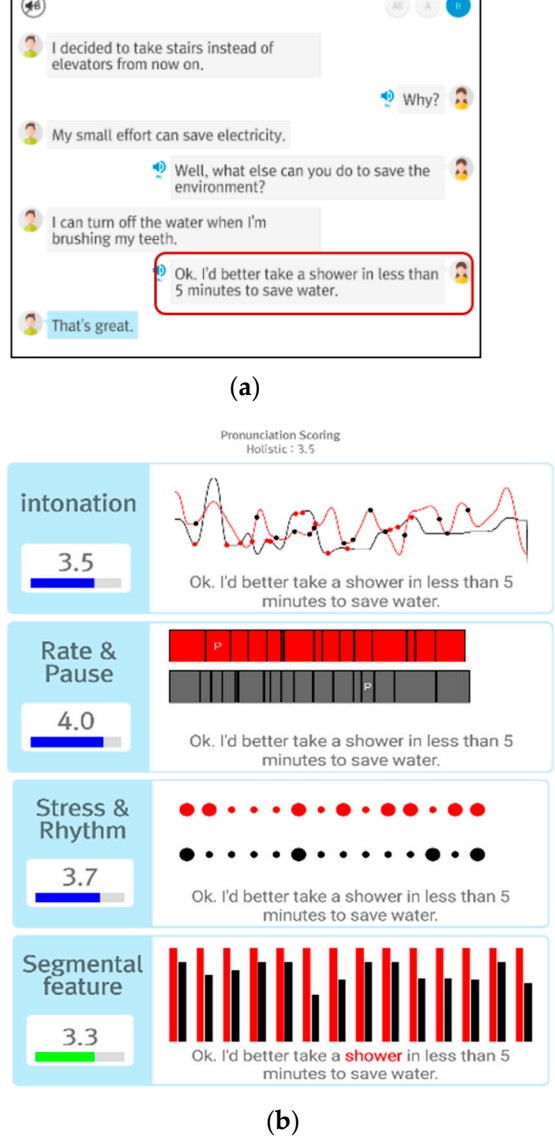

**Figure 7.** Example of dialogue exercise and fluency scores of the learner and the native speaker with the proposed method: (**a**) example of a role-play dialogue exercise, (**b**) fluency scores feedback.

### 4.2. Proficiency Evaluation Test

### 4.2.1. Speech Database

We also performed the proficiency evaluation test using the rhythm and stress scores with other fluency features. A speech dataset was selected from the English read speech dataset read by non-native and native speakers for the spoken proficiency assessment. The dataset is a corpus of English speech sounds spoken by Koreans and 7 American English native speakers (references) for experimental phonetics, phonology, and English education, and is designed to see Korean speakers' intonation and rhythmic patterns in English connected speech and the errors which Korean speakers are apt to make in pronunciation of segments. Each utterance was scored by human expert raters on a scale of 1 to 5. In this study, the gender and spoken language proficiency levels were evenly distributed among the speakers. Table 3 shows scripts samples. The speech dataset comprised 100 non-native speakers, and for each speaker, 80 sentences were used for training and another 20 sentences, not included in the training dataset, were used for testing.

**Table 3.** Samples of the scripts.

| No. | Sentence |
| --- | --- |
| 1 | My pet bird sleeps in the cage. |
| 2 | I eat fruits when I am hungry. |
| 3 | Miss Henry drank a cup of coffee. |
| . . . | |
| 100 | They suspect that the suspect killed Ted. |

For speech conversion and augmentation, an additional 7 American English native speakers (3 males and 4 females), and for each speaker, 100 sentences, were used, and frame alignment of the dataset was not performed. We used the WORLD package [49] to perform speech analysis. The sampling rate of all speech signals reported in this paper was 16 kHz. The frame shift length was 5 ms and the number of fast Fourier transform (FFT) points was 1024. For each extracted spectral sequence, 80 Mel-cepstral coefficients (MCEPs) were derived.

### 4.2.2. Human Expert Rater

Each spoken English sentence uttered by non-native learners was annotated by four human expert raters who have English teaching experience or are currently English teachers. Each non-native utterance was rated for five proficiency area scores: holistic impression of proficiency, intonation, stress and rhythm, speech rate and pause, and segmental accuracy. In addition, each proficiency score was measured on a fluency level scale of 1–5. A holistic score for each utterance is calculated as an average of all proficiency scores and used for proficiency evaluation in this paper. Table 4 shows a mean of the correlation between human expert raters' holistic scores.

**Table 4.** Inter-rater correlation.

| Rater | 2 | 3 | 4 |
| --- | --- | --- | --- |
| 1 | 0.79 | 0.75 | 0.80 |
| 2 | - | 0.81 | 0.83 |
| 3 | - | - | 0.80 |

### 4.2.3. Data Augmentation

The proposed VAE-based speech conversion model consisted of a content encoder, style encoders, and a joint decoder. The content encoder comprised two dilated convolutional layers and a gated recurrent unit (GRU) based on a recurrent neural network. In order to remove the speech style information, all convolutional layers were followed by instance normalization (IN) [50]. The style encoder comprised a global average pooling layer,

3-layer multi-layer perceptron (MLP), and a fully connected layer. In the style encoder, IN was not used because it removes the original feature mean and variance that represent speech style information. Then, content and style factors were fed into the decoder to reconstruct or convert the speech. The decoder comprised two dilated convolutional layers and the recurrent neural network-based GRU. All convolutional layers were used with an Adaptive Instance Normalization layer generated by the MLP from the style factor [50].

$$AdaIN(z, s) = \sigma(s) \left( \frac{z - \mu(z)}{\sigma(z)} \right) + \mu(s), \tag{7}$$

where $z$ is the activation of the previous convolutional layer, and $\mu(.)$ and $\sigma(.)$ denote the mean and variance, respectively.

Figure 8 shows an example of Mel-spectrograms obtained by the proposed method. Comparing the decoding results, we confirmed that the proposed method reconstructs and converts the spectral features efficiently.

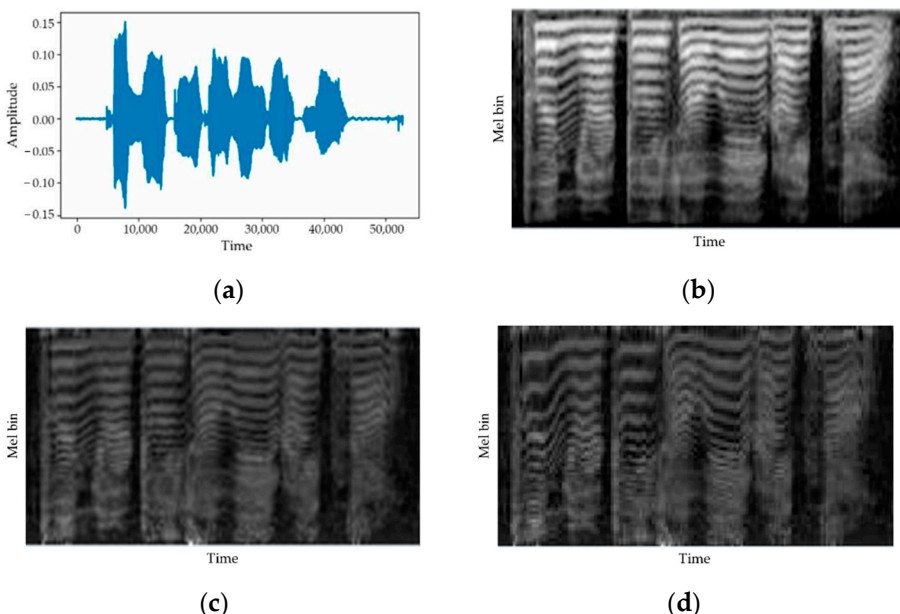

**Figure 8.** Waveform and Mel-spectrograms. (**a**) Waveform of the input signal, (**b**) Mel-spectrogram of the input signal, (**c**) reconstructed Mel-spectrogram, and (**d**) converted Mel-spectrogram.

We performed the perception test to compare the sound quality and speaker similarity of converted speech between the proposed VAE-based speech conversion method and the conventional conditional VAE-based speech conversion (CVAE-SC) method [29], which is one of the most common speech conversion methods. We conducted an AB test and an ABX test. "A" and "B" were outputs from the proposed method and the CVAE, and "X" was a real speech sample. To eliminate bias in the order, "A" and "B" were presented in random orders. In the AB test, each listener was presented with "A" and "B" audios at a time, and was asked to select "A", "B", or "fair" by considering both speech naturalness and intelligibility. In the ABX test, each listener was presented with two audios and a reference audio "X", and then, was asked to select a preferred audio or "fair" by considering the one closer to the reference. We used 24 utterance pairs for the AB test and another 24 utterance pairs, not included in the AB test, for the ABX test. The number of listeners was 20. Figure 9 shows the results, and we confirmed that the proposed method outperforms the baseline in both sound quality and speaker similarity terms.

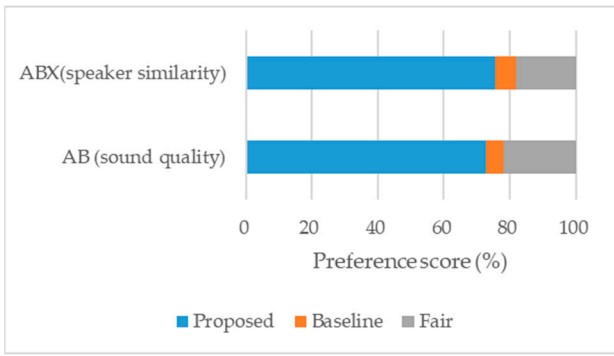

**Figure 9.** Results of the AB test and the ABX test.

We also performed the speech recognition test to validate that the spectral features were converted meaningfully using the English read speech dataset. We used the ESP-net [51] for an end-to-end ASR system. We trained the AM using only the training dataset ("Train database only" in Table 5) and evaluated the test dataset, and we compared the recognition results to those obtained by evaluating the same test dataset using the AM trained with the augmented dataset ("Augmentation" in Table 5). Table 5 shows the word error rate (WER) results. For comparison, SpecAugment [21], speed perturbation method [20], and CVAE-SC were used as a reference. As shown in Table 5, we confirmed that the data augmentation with the proposed method improves the speech recognition accuracy for all proficiency score levels compared to a method employing conventional AM and the other augmentation methods. By sampling different style factors, the proposed speech conversion method is able to generate diverse outputs, but the computational complexity is higher than that of other methods.

**Table 5.** Speech augmentation and word error rate (%) results.

| Proficiency Level | Applied Method | 1 | 2 | 3 | 4 | 5 |
|---|---|---|---|---|---|---|
| Train database only | - | 57.3 | 53.4 | 30.1 | 23.4 | 22.7 |
| Augmentation | SpecAugment | 52.1 | 45.4 | 27.2 | 21.9 | 20.6 |
| | Speed perturbation | 51.3 | 44.9 | 27.0 | 21.8 | 20.7 |
| | CVAE-SC | 49.3 | 43.1 | 26.0 | 21.5 | 20.4 |
| | Proposed method | 40.9 | 37.7 | 24.5 | 20.5 | 19.1 |

### 4.2.4. Features for Proficiency Scoring

All features for proficiency scoring are computed based on the time-aligned phone sequence and its time information [11,12,14]. Table 6 shows the proficiency scoring feature list used to train the automatic proficiency scoring models in this work.

**Table 6.** Features for the proficiency scoring modeling.

| Feature Name | Description |
|---|---|
| Genie_pron | Pronunciation score |
| SLLR | Sentence-level log-likelihood ratio |
| amscore0/amscore1 | Acoustic model (AM) score/anti-model-based AM score |
| Uttsegdur | Duration of entire transcribed segment but without inter-utterance pauses |
| Globsegdur | Duration of entire transcribed segment, including all pauses |
| wpsec/wpsecutt | Speech articulation rate/words per second in utterance |
| Silpwd/Silpsec | Number of silences per word/second |
| Numsil | Number of silences |
| Silmean/Silmeandev/Silstddev | Mean/mean deviation/standard deviation of silence duration in second |
| Longpfreq/Longpwd | Frequency/number of long pauses per word (0.495 s $\leq$ duration) |
| Longpmn/Longpmeandev/Longpstddev | Mean/mean deviation/standard deviation of long pauses |
| Wdpchk/Secpchk | Average speech chunk length in words/seconds |
| Wdpchkmeandev/Secpchkmeandev | Mean deviation of chunks in words/seconds |
| Repfeq | Number of repetitions divided by number of words |

**Table 6.** *Cont.*

| Feature Name | Description |
| --- | --- |
| Tpsec | Types (unique words) per second |
| Tpsecutt | Types normalized by speech duration |
| Wdstress/Uttstress | Word/sentence stress score |
| Rhymean/Rhystddev | Mean/standard deviation of rhythm score |
| FlucMean/Flucstddev | Mean/standard deviation of the range of fluctuation |
| propV | Vocalic segment duration ratio in sentence |
| deltaV/deltaC | Standard deviation of vocalic/consonantal segment duration |
| varcoV/varcoC | Standard deviation of vocalic/consonantal segment duration normalized by speech rate |
| Genie_amscoreK0/Genie_amscoreK1 | AM score/anti-model AM score reflecting Korean pronunciation characteristics of English |
| Numdff | Number of disfluencies |
| Dpsec | Disfluencies per second |

### 4.2.5. Proficiency Scoring Model

We used two modeling methods: (1) multiple linear regression (MLR) and (2) deep neural network, to train scoring models with high agreement with human expert raters. MLR is simple and has been used for a long time for automatic proficiency scoring purposes. Based on the MLR scoring model, the proficiency score is computed as follows:

$$Score = \sum_i \alpha_i \cdot f_i + \beta, \tag{8}$$

where $i$ is the index of each feature, $\alpha_i$ is the weight associated with each scoring feature $f_i$, and $\beta$ is a constant intercept.

We also used a neural network to train the proficiency scoring model nonlinearly and more accurately. The neural network comprised a convolutional layer with 1 hidden layer and 3 hidden units and a fully connected layer. Given 41 features, the neural network trains the proficiency scoring model.

### 4.2.6. Proficiency Evaluation Results

In order to validate that the proposed automatic proficiency evaluation system measured the proficiency scores effectively and meaningfully, we computed and compared a Pearson's correlation coefficient between the proficiency scores of the proposed system and those of human raters. The Pearson's correlation coefficient is a commonly used metric for evaluating the performance of proficiency assessment methods [52–54]. Tables 7 and 8 show the proficiency evaluation results obtained by the proposed method with and without data augmentation. For comparison, the range of correlation coefficients of the inter-rater scores ("Human" in Tables 7 and 8) were used as a reference. As shown in Tables 7 and 8, we confirmed that the proposed automatic proficiency evaluation method measures proficiency scores efficiently for all proficiency area scores. In addition, we confirmed that data augmentation for AM training with the proposed speech conversion method improves the averaged correlation performance for all proficiency area scores compared to the method employing conventional AM trained without data augmentation. By automatically evaluating the proficiency of the L2 speaker's utterance, the proposed proficiency scoring system is able to perform fast and consistent evaluation in various environments.

**Table 7.** Correlation between human rater and proposed proficiency scoring system without data augmentation.

| Rater | Human | MLR | Neural Network |
| --- | --- | --- | --- |
| Holistic | 0.68~0.79 | 0.78 | 0.82 |
| Intonation | 0.64~0.72 | 0.73 | 0.77 |
| Stress and rhythm | 0.71~0.74 | 0.75 | 0.78 |
| Speech rate and pause | 0.71~0.75 | 0.75 | 0.77 |
| Segmental features | 0.59~0.67 | 0.69 | 0.73 |

**Table 8.** Correlation between human rater and proposed proficiency scoring system with data augmentation.

| Rater | Human | MLR | Neural Network |
|---|---|---|---|
| Holistic | 0.68~0.79 | 0.83 | 0.84 |
| Intonation | 0.64~0.72 | 0.78 | 0.80 |
| Stress and rhythm | 0.71~0.74 | 0.78 | 0.79 |
| Speech rate and pause | 0.71~0.75 | 0.81 | 0.82 |
| Segmental features | 0.59~0.67 | 0.73 | 0.76 |

## 5. Conclusions and Future Work

We proposed an automatic proficiency evaluation method for L2 learners in spoken English. In the proposed method, we augmented the training dataset using the VAE-based speech conversion model and trained the acoustic model (AM) with an augmented training dataset to improve the speech recognition accuracy and time-alignment performance for non-native speakers. After recognizing the speech uttered by the learner, the proposed method measured various fluency features and evaluated the proficiency. In order to compute the stress and rhythm scores even when the phonemic sequence errors occur in the learner's speech, the proposed method aligned the phonemic sequences of the spoken English sentences by using the DTW, and then computed the error-tagged stress patterns and the stress and rhythm scores. In computer experiments with the English read speech dataset, we showed that the proposed method effectively computed the error-tagged stress patterns, stress scores, and rhythm scores. Moreover, we showed that the proposed method efficiently measured proficiency scores and improved the averaged correlation between human expert raters and the proposed method for all proficiency areas compared to the method employing conventional AM trained without data augmentation.

The proposed method can also be used for most signal processing and generation problems, such as sound conversion between instruments or generation of various images. However, the current style conversion framework has a limitation that the conversion model learns the domain-level style factors and generates the converted speech signal rather than diverse pronunciation styles of multiple speakers included in each domain. In order to learn more meaningful and diverse style factors and perform many-to-many speech conversion, we plan to address the issues of automatic speaker label estimation and expansion to each speaker-specific style encoder in the future work.

**Author Contributions:** Conceptualization, methodology, validation, formal analysis, writing—original draft preparation, and writing—review and editing, Y.K.L.; supervision and project administration, J.G.P. All authors have read and agreed to the published version of the manuscript.

**Funding:** This work was supported by an Electronics and Telecommunications Research Institute (ETRI) grant funded by the Korean government (21ZS1100, Core Technology Research for Self-Improving Integrated Artificial Intelligence System), and by the Institute of Information & Communications Technology Planning & Evaluation (IITP) grant funded by the Korean Government (MSIT) (2019-0-2019-0-00004, Development of semi-supervised learning language intelligence technology and Korean tutoring service for foreigners).

**Institutional Review Board Statement:** Not applicable.

**Informed Consent Statement:** Not applicable.

**Data Availability Statement:** Not applicable.

**Conflicts of Interest:** The authors declare no conflict of interest.

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
