# Peer review of "Multimodal Unsupervised Speech Translation for Recognizing and Evaluating Second Language Speech"

_applsci, doi:10.3390/app11062642_

Round 1

Reviewer 1 Report

This manuscript describes an automatic proficiency evaluation and speech recognition for second-language speech. The whole research is well-written, and the structure of the manuscript is very clear. My main comment is about the lack of necessary information to verify/repeat the study: 1) could you provide more descriptions regarding the features of native speakers? Native English speaker is a very big concept, more features would be necessary for readers to understand the strenghs and limitations of the study result. 2) could you also provide more descriptions regarding the features of the L2 speakers as well? 3) could you briefly describe the limitations and generalizability of the study methods/results? 4) I failed to find any extra materials / online repositories where I can find the corresponding data, etc. Could you please indicate where the materials can be accessed? if it is not possible, could you please provide an alternative way or explain why?

Author Response

Reviewer#1, Concern # 1 and 2: This manuscript describes an automatic proficiency evaluation and speech recognition for second-language speech. The whole research is well-written, and the structure of the manuscript is very clear. My main comment is about the lack of necessary information to verify/repeat the study:

1) could you provide more descriptions regarding the features of native speakers? Native English speaker is a very big concept, more features would be necessary for readers to understand the strenghs and limitations of the study result.

2) could you also provide more descriptions regarding the features of the L2 speakers as well?

Author response: Thank you for your very helpful comments. We agree that more feature descriptions about native English speakers and L2 speakers would be necessary for readers. The dataset used in the paper is a corpus of the English read speech dataset read by nonnative speakers (Korean, L2 English learner) and American English native speakers for reference (teacher). It was designed to see Korean speakers’ intonation and rhythmic patterns in English and the errors which Korean speakers are apt to make in pronunciation of segments. Each utterance was scored by human expert raters on a proficiency level of 1 to 5.

We updated the manuscript by adding a description of the dataset used in this paper and adding a new script sample table. The updated description and table can be shown on page 10.

Reviewer#1, Concern # 3: could you briefly describe the limitations and generalizability of the study methods/results?

Author response: We agree that it is important to describe the limitations and generalizability of the study. The proposed speech conversion method generates diverse converted speech signals by extracting and converting some speech characteristics (style factors) for each domain. However, each domain includes multiple speakers, and accordingly, the conversion model learns the domain-level style factors rather than detailed styles such as pronunciation styles of individual speakers. If the models are trained for multiple speakers, the speech conversion model can learn more meaningful and diverse styles and conversion can be performed more effectively. Also, the proposed method can perform many-to-many speech conversion for multiple speakers by adding each speaker’s style encoder. Therefore, we plan to address the issues of automatic speaker label estimation and many-to-many speech conversion in the future work.

We updated the manuscript by adding a description of the limitations and future work for generalizability of the study to the conclusion. The updated conclusion can be shown on page 15.

Reviewer#1, Concern # 4: I failed to find any extra materials / online repositories where I can find the corresponding data, etc. Could you please indicate where the materials can be accessed? if it is not possible, could you please provide an alternative way or explain why?

Author response: Thank you for your comment. The corresponding dataset is currently being used in commercial language learning system for untact English learning. According to the company’s data and information security policy, we may only provide approved information. We are currently in the process of accepting an application for registration for releasing data design description and samples, and after the approval process is completed, we will provide corresponding materials on github or online paper.

Instead, we updated the manuscript by adding a description of the dataset and adding script samples by a new table. The updated description and table can be shown on page 10. 

Reviewer 2 Report

This paper clearly presents the solution of speech conversion and evaluation on nonnative English speech for both content and styles. 

In section 3.2, the description of the conversion model is not very clear. In the unsupervised learning, how can we ensure the content factor and style factor to learn towards to targets? Can you show more details of the loss functions and explain why they work?

In section 4, besides evaluating the system on the proficiency scores and WERs, can you provide more direct quantitative measures on the style factors?

The overall application and impact of the proposed approach need to be addressed further based on the results.

Author Response

Reviewer#2, Concern # 1: This paper clearly presents the solution of speech conversion and evaluation on nonnative English speech for both content and styles.

In section 3.2, the description of the conversion model is not very clear. In the unsupervised learning, how can we ensure the content factor and style factor to learn towards to targets? Can you show more details of the loss functions and explain why they work?

Author response:  Thank you for your very helpful comments. In the proposed speech conversion model, we assume that each spectral feature is decomposed into a content factor and style factor. The content factor is a speaker-independent information we want to maintain and it is a phoneme in speech. In order to compute content factors, one content encoder is shared across all domains. Also, all convolutional layers of the content encoder were followed by instance normalization to remove the speech style information. By using a shared content encoder for all domains and applying instance normalization to all convolutional layers, the content encoder learns domain-independent content information. The style encoder computes the domain-specific style factors (information we want to change) for each domain. In the style encoders, instance normalization is not applied, because it removes the speech style information.

In order to ensure the content encoder and style encoders learn content and style factors, we compute a semi-cycle loss. For example, a content reconstruction loss (Equation 4) encourages the content encoder to learn the semantic content information of input spectral features. The content reconstruction loss for source speech signal is computed as follows:

  1. Compute the content factor with input source spectral features (c in equation 4)
  2. Compute the target style factor with input target spectral features (st in equation 4)
  3. Generate the converted signal with computed source content factor (1) and target style factor (2)
  4. Compute the content factor again with converted signal (3)
  5. Compute the content reconstruction loss (4 – 1, equation 4)

The content reconstruction loss for target speech is similarly computed.

  1. Compute the content factor with input target spectral features
  2. Compute the source style factor with input source spectral features
  3. Generate the converted signal with computed target content factor (1) and source style factor (2)
  4. Compute the content factor again with converted signal (3)
  5. Compute the content reconstruction loss for target speech (4 – 1)

Two content reconstruction losses (for source and target) are used to compute a total loss. This corresponds to  in Equation 6. The style reconstruction losses for source and target speech are also similarly computed, and they encourage style encoders to extract domain-specific style information.

We updated the manuscript by adding the description of the conversion model. The updated description can be shown on pages 8 and 9.

Reviewer#2, Concern # 2: In section 4, besides evaluating the system on the proficiency scores and WERs, can you provide more direct quantitative measures on the style factors? The overall application and impact of the proposed approach need to be addressed further based on the results.

Author response:  Thank you for your suggestion. We have incorporated your comments by performing the additional perception tests (AB test and ABX test) using the converted outputs and comparing them.

We updated the manuscript by adding the perception test results to the experimental results. The updated experimental results are shown on page 12, Figure 9.

Reviewer 3 Report

-English should be corrected
-please add colorful picture of measurements (optionally);;; + arrows what is what
-please add block diagram of the proposed research step by step ;;; what is the result of paper?;;; Fig. 1?
-please add photo/photos of application of the proposed research ;;;;
-please add sentences about future analysis;;;
-Figures should have better quality;;;;
-Please add labels to axes (Figures);;;;
-please add arrows to photos what is what;;;
-formulas and fonts should be formatted;;;;
-references should be 2018-2021 Web of Science about 50% or more ;; 30 at least
-Please compare with other methods, justify. Advantages or Disadvantages different methods
-Conclusion: point out what are you done;;;;

Author Response

Response to Reviewer 3 Comments

Point 1: English should be corrected 

Response 1: Thank you for your helpful comments. We updated the manuscript by checking and correcting the English spelling and grammar.

Point 2: please add colorful picture of measurements (optionally);;; + arrows what is what

Response 2: Thank you for your suggestion. We have incorporated your comments by updating the figures used in the document and adding the description of what each arrow means in Figure 6 to make the figure more clearly.

Point 3: please add block diagram of the proposed research step by step ;;; what is the result of paper?;;; Fig. 1? 

Response 3: The proposed research is an automatic proficiency evaluation method that recognizes the speech uttered by the L2 English speaker, computes various fluency features, and evaluates the proficiency of the speaker’s spoken English. Block diagram of the proposed research is Figure 3, and the experimental results of the paper can be shown on Section 4.

Point 4: please add photo/photos of application of the proposed research ;;;; ʉ۬

Response 4: Thank you for your comment. The proposed automatic proficiency evaluation model is currently being used in English learning system for L2 learners, and examples (photos of application) of the system can be shown on Figures 2 and 7.

Point 5: please add sentences about future analysis;;; 

Response 5: We agree that it is important to describe the future work of the study. The proposed method can perform many-to-many speech conversion for multiple speakers by adding each speaker’s style encoder. In addition, if the models are trained for multiple speakers, the speech conversion model can learn more meaningful and diverse styles and conversion can be performed more effectively. Therefore, we plan to address the issues of automatic speaker label estimation and many-to-many speech conversion in the future work.

We updated the manuscript by adding a description of the future work of the study to the conclusion. The updated conclusion can be shown on page 15.

Point 6: Figures should have better quality;;;; ʉ۬

Response 6: Thank you for your comment. We updated the manuscript by improving the quality of all figures (figure 1 and figure 3 to 6).

Point 7: Please add labels to axes (Figures);;;;

Response 7: Thank you for catching this. We have added labels to axes in Figure 8.

Point 8: please add arrows to photos what is what;;; 

Response 8: Thank you for your comment. We have incorporated your comments by adding the description of what each arrow means in Figure 6 to make the figure more clearly.

Point 9: formulas and fonts should be formatted;;;; 

Response 9: Thank you so much for catching this. We have checked and modified the fonts of all formulas and figures to follow the format.

Point 10: references should be 2018-2021 Web of Science about 50% or more ;; 30 at least

Response 10: We agree that more recent works should be referred. We have updated the manuscript by adding some recent (2018-2020) works to the references. The updated references can be shown on pages 15, 16, and 17.

Point 11: Please compare with other methods, justify. Advantages or Disadvantages different methods

Response 11: Thank you for your suggestion. In the introduction, we compared the proposed fluency scoring and speech conversion method with conventional pattern comparison scoring methods and generative model-based conversion methods. Brief descriptions of prior works can be shown on pages 2 and 3. In order to add advantages or disadvantages of the proposed method, we have performed an additional data augmentation experiment, and we also have added some discussion of the data augmentation experiment and proficiency evaluation results. The updated experimental results and descriptions are shown on following pages:

Experimental results: page 13

Discussion: pages 13 and 14

Point 12: Conclusion: point out what are you done;;;;

Response 12: Thank you for your comment. We have updated the conclusion by modifying the conclusion and adding a description of future work. The updated conclusion can be shown on page 15.

Round 2

Reviewer 1 Report

no more extra comments.

Author Response

Reviewer's comment: no more extra comments.

Author response: Thank you very much for your efforts to improve this manuscript.

Reviewer 3 Report

-Literature of paper is old. Correct it. You should show new knowledge,
-references should be 2015-2021 Web of Science about 50% or more ;; 30 at least
-Please compare with other methods, justify. Advantages or Disadvantages ;;; 

for example 

1) 
Recognition of acoustic signals of induction motor using Fft, Smofs-10 and LSVM, Eksploatacja i Niezawodnosc – Maintenance and Reliability,
2015, 17 (4), pp. 569-574. https://doi.org/10.17531/ein.2015.4.12

-is there possibility to use the proposed methods for other problems for example fault diagnosis of motors/engines?

Author Response

Point 1: Literature of paper is old. Correct it. You should show new knowledge, references should be 2015-2021 Web of Science about 50% or more ;; 30 at least 

Response 1: Thank you for your comments. We have updated the manuscript by adding some 2015-2021 Web of Science to the references. The updated references can be shown on pages 16 and 17.

Point 2: Please compare with other methods, justify. Advantages or Disadvantages different methods

for example

1)

Recognition of acoustic signals of induction motor using Fft, Smofs-10 and LSVM, Eksploatacja i Niezawodnosc – Maintenance and Reliability,

2015, 17 (4), pp. 569-574. https://doi.org/10.17531/ein.2015.4.12

-is there possibility to use the proposed methods for other problems for example fault diagnosis of motors/engines?

Response 2: Thank you for your suggestion. The proposed method corresponds to the signal processing and domain transformation problem. The proposed method can be used for most signal processing, generation, conversion, or recognition problems, using signals (e.g. speech, image, musical instrument sounds, physiological signal, machine-related sounds, etc.). We have added to the conclusion a discussion of the applicability of the proposed method to other problems. The updated description is shown on page 15.
